# Are Embedded Potatoes Still Vegetables?
# On the Limitations of WordNet Embeddings for Lexical Semantics

**Xuyou Cheng**
xc341@cantab.ac.uk

**Michael Schlichtkrull**
University of Cambridge
mss84@cam.ac.uk

**Guy Emerson**
University of Cambridge
gate2@cam.ac.uk

|        | Similarity | Analogy | POS    | NER    |
|--------|------------|---------|--------|--------|
| MRR    | $-0.749$   | $-0.539$| $-0.645$| $-0.757$|
| H@1    | $-0.843$   | $-0.650$| $-0.747$| $-0.746$|
| H@10   | $-0.355$   | $-0.138$| $-0.220$| $-0.705$|

Table 1: Pearson correlation between link prediction metrics and other evaluation tasks, for the results in Table 2. For all metrics, higher is better, but the link prediction metrics are anticorrelated with the other tasks.

## Abstract

Knowledge Base Embedding (KBE) models have been widely used to encode structured information from knowledge bases, including WordNet. However, the existing literature has predominantly focused on link prediction as the evaluation task, often neglecting exploration of the models' semantic capabilities. In this paper, we investigate the potential disconnect between the performance of KBE models of WordNet on link prediction and their ability to encode semantic information, highlighting the limitations of current evaluation protocols. Our findings reveal that some top-performing KBE models on the WN18RR benchmark exhibit subpar results on two semantic tasks and two downstream tasks. These results demonstrate the inadequacy of link prediction benchmarks for evaluating the semantic capabilities of KBE models, suggesting the need for a more targeted assessment approach.

## 1 Introduction

Knowledge Base Embedding (KBE) models aim to encode the structured information contained in knowledge bases as vectors. These models are often claimed to possess rich semantic information, and have been applied to various downstream semantic tasks, such as entity typing (Moon et al., 2017; Zhao et al., 2020), entity alignment (Sun et al., 2020), rule mining (Yang et al., 2015; Chen et al., 2020), and conceptual clustering (Gad-Elrab et al., 2020). If the knowledge base encodes semantic relations (hypernymy, meronymy, and so on), a KBE model should represent these relations. The assumption is that this knowledge should be helpful for more general semantic modelling. A further assumption is that evaluating on the training task, i.e. relational link prediction, is suitable as a proxy for other tasks. Our aim is to test these implicit assumptions.

We focus on models of the Princeton WordNet (Miller, 1995). Typically, KBEs are evaluated on link prediction for an artificially selected subgraph. For instance, the WN18 benchmark (Bordes et al., 2014) is created from WordNet by excluding rare synsets and relations, and randomly splitting the data into training and test sets. Further, following issues with test set leakage (Toutanova and Chen, 2015; Dettmers et al., 2018), the subsequent revision WN18RR meticulously filters the splits, removing e.g. explicit inverse relations.

While link prediction may be a natural task for datasets such as FreeBase, where incompleteness is a major problem (Min et al., 2013), the Princeton WordNet is intended to have complete annotations of lexical relations. Indeed, it has often been used as the basis for developing WordNets for languages other than English (Bond and Foster, 2013). The careful filtering of WN18RR might result in a fairer benchmark for link prediction, but this evaluation setup deviates from the objective of learning a semantically valuable model.

Some of the semantic information in WordNet is implicit. For example, *hypernymy* (i.e., the "is a" relation) is transitive, but antonymy (the "opposite of" relation) is not; nevertheless, they are both encoded as synsets having arcs to their direct hypernyms or antonyms (Miller, 1995). That is, while *hypernym(potato, solanaceous vegetable)* and *hypernym(solanaceous vegetable, vegetable)* are relations in WordNet, *hypernym(potato, vegetable)* is not. It is supposed to be inferred by the

user's knowledge of hypernymy. However, link prediction models are neither trained nor tested for this; rather, they are trained to *exclude* indirect hypernym-relations like *hypernym(potato, vegetable)*, as they are not present in the training or the test splits. Indeed, to a KBE model, hypernymy and antonymy look very similar.

Putting together the heavy filtering of common datasets and the indirectness of some information in WordNet, it is unclear the degree to which KBE models trained on these datasets contain the semantic information WordNet is designed to include. Indeed, we find that well-performing KBE models on WN18RR are *anticorrelated* with performance on downstream semantic tasks: the best models of WN18RR perform the worst on semantic benchmarks (see Table 1). This aligns with findings for other types of embedding, where a discrepancy in model performance between intrinsic and extrinsic metrics has been observed (see §3).

We provide evidence for two related claims: I) performance on link prediction is not predictive of semantic capability; and II) models developed for link prediction fail to capture the semantics of WordNet. Specifically, we show that models with the best performance on the WN18RR benchmark exhibit poor performance on relatively simple semantic tasks (as seen in Table 1, and discussed in §5). We further show that well-performing KBE models fail to properly encode the most basic semantic relationship in WordNet, i.e. hypernymy (as discussed in §6). Our findings emphasize the need for careful design of both evaluation benchmarks and model architectures: evaluation must target desired functionality, and models must be theoretically capable of that functionality.

## 2 KBE Models

A Knowledge Base consists of triples $(s, r, o)$, denoting the subject node $s$ being connected by the relation $r$ to the object node $o$. KBE models aim to learn an embedding for each node and relation, so that it can output higher scores for valid triplets and lower scores for invalid ones.

In this paper, we compare several common models, including TransE (Bordes et al., 2013), DistMult (Yang et al., 2015) and MuRP (Balažević et al., 2019), as well as a function-based model, FuncE, adapted from Chen (2021). We also discuss GNN-based and distributional approaches.

TransE embeds both nodes and relations as real vectors, with scores defined by translation: $-|v_s + v_r - v_o|^2$. DistMult also uses real vectors, but instead defines scores by componentwise multiplication: $\Sigma_i v_{s,i} v_{r,i} v_{o,i}$.

Some KBE models have been designed to meet specific geometric and linguistic criteria (Vendrov et al., 2016; Nickel and Kiela, 2017). For the multirelational setting, Balažević et al. (2019) introduce MuRP, which encodes relations as Möbius transformations, with scores defined using the distance between transformed points. Allen et al. (2021) analyse the performance of different scoring functions on different categories of relations. Their findings underscore that models should be selected for specific predictive needs: DistMult performs better for some relation types, MuRP for others. Similarly, Chen (2021) proposed to use different scoring functions for different WordNet relations, motivated by theoretical work on Functional Distributional Semantics (Emerson, 2018). Building on this, we introduce FuncE, which embeds a node as a real-valued function $f : \mathbb{R}^n \to [0, 1]$, and uses a different scoring function for different categories of relation, according to linguistic criteria: hypernym-based, synonym-based, or other (see Appendix A).

Such a function can be interpreted as the membership function for fuzzy region (Zadeh, 1965). In distributional semantics, Erk (2009b,a) has argued in favour of such representations (for a survey, see: Emerson, 2020). Membership functions can naturally represent tree hierarchies (like hyperbolic embeddings) but can also naturally represent multiple inheritance hierarchies. Using a real-valued space makes it easier to switch scoring functions for different types of relation.

More precisely, $f$ is a hypernym of $g$ if $f(x) \geq g(x)$ for all $x$. For simplicity, we embed nodes as logistic regression classifiers, defined by a weight vector $v$ and bias $b$, so $f(x) = \text{sigmoid}(v.x + b)$. This allows hypernymy to be calculated easily: restricting the input to the unit sphere, $f$ is a hypernym of $g$ if $b_f - b_g - |v_f - v_g| \geq 0$. We can use this expression as a score for the hypernymy relation. Numerically, this is the same hypernymy condition as for ball embeddings (Dong et al., 2018), if we take $v$ to be the centre of the ball and $b$ to be the radius. For synonym-based relations, we use the DistMult score function, and for other relations, we use the TransE score function.

We further experiment with recent works that apply Graph Neural Networks (GNNs) as node encoders, which can potentially leverage the structural context of each node (Schlichtkrull et al., 2018). In particular, we consider KBGAT (Nathani et al., 2019) and rGAT (Chen et al., 2021), both of which were found to have high performance on link prediction for WN18RR.

Finally, some work in distributional semantics has compared models trained on corpora against models trained on linguistic resources like WordNet. Faruqui and Dyer (2015) create sparse binary vectors, based on manually defined features, and find that they are competitive with distributional vectors on various evaluation datasets. Kutuzov et al. (2019) learn synset embeddings from WordNet by utilizing path-length based similarity measures. Saedi et al. (2018) present Wnet2vec, which generates word embeddings following the intuition that semantically similar nodes are connected by more paths and shorter paths. They show that the resulting embeddings outperform several distributional methods on word similarity metrics. We consider Wnet2vec as an additional baseline.

## 3  Related Research on Evaluation

A growing body of research advocates a reassessment of the evaluation protocol for KBE models, even if link prediction is taken as the ultimate goal. Wang et al. (2019) and Akrami et al. (2018, 2020) argue that current benchmarks contain too many trivial and too few difficult predictions, advocating for more realistic evaluation. Tiwari et al. (2021) discuss how random splits alter the graph topology from the original knowledge base, and show that correcting for this allows simple translation-based models to outperform more complicated ones. Finally, Kadlec et al. (2017) and Ruffinelli et al. (2020) show that performance on link prediction benchmarks is strongly determined by hyperparameters and training mechanisms.

Going beyond link prediction, Jain et al. (2021) challenge the prevalent belief that KBE models can properly capture the semantics of KB entities and relations. By evaluating common KBE models on entity typing and entity clustering benchmarks, the authors discovered that their results were no better than those achieved by simple statistical models. Similarly, Rim et al. (2021) designed tests for specific capabilities of KBE models, such as capturing symmetric relations or hierarchical relations. They

found that performance on link benchmarks did not reflect performance on their tests. Together, these studies suggest that link prediction is a narrow evaluation metric, even when considering tasks defined on the knowledge base itself. Our work differs from these studies as we primarily focus on external evaluation datasets, investigating the limitations of WordNet embeddings in representing lexical semantics. Specifically, we apply WordNet KBE models to semantic evaluation datasets and downstream tasks.

Differences between training-related and downstream measures of performance have been found for several NLP tasks. Chiu et al. (2016), Qiu et al. (2018), and Torregrossa et al. (2020) found that performance on common benchmarks for word embeddings do not correlate well with downstream performance. Similar results have been demonstrated especially for intrinsic measurements of bias and fairness, which often struggle to adequately capture downstream failures (Goldfarb-Tarrant et al., 2021; Cao et al., 2022).

## 4  Experimental Setup

To demonstrate how performance on link prediction benchmarks is inadequate for assessing semantic capabilities, we evaluate common link prediction models on several semantic tasks. We use two tasks directly testing semantic understanding (word similarity, word analogy), and two downstream tasks using the embeddings (POS-tagging, NER). These tasks serve as representative examples of challenges that semantic embedding models should be able to tackle effectively.

### 4.1  Word Similarity

A common approach for evaluating semantic models is to assess their ability to predict how similar words are. Typically, this is done by comparing model predictions with human-annotated similarity scores for selected word pairs. Model performance is then measured with Spearman correlation. For models that represent words as vectors, cosine similarity is typically employed to measure word similarity. In our analysis, we use the SimLex999 benchmark (Hill et al., 2015a), which includes 999 pairs of common words spanning three parts of speech (nouns, verbs, and adjectives). Each sample in the benchmark consists of a pair of words along with a ground-truth similarity score assigned by crowd-sourcing (e.g. "coast shore 9.00").

| | WN18RR | | | Similarity | Analogy | POS-tagging | NER |
|---|---|---|---|---|---|---|---|
| | MRR | @10 | @1 | Spearman $\rho$ | @10 | acc. | f1 score |
| TransE | 34.49 | 50.94 | 25.27 | 48.60 | 31.98 | 76.53 | 49.18 |
| DistMult | 42.47 | 47.40 | 39.74 | _28.76_ | _11.60_ | 67.17 | 48.43 |
| FuncE | _25.87_ | _43.66_ | _16.78_ | **51.19** | **35.30** | **77.17** | **54.54** |
| MuRP | 48.85 | 58.64 | 43.40 | 38.16 | 30.59 | 67.87 | 43.89 |
| KBGAT | 44.45 | 58.31 | 36.72 | 45.05 | 31.56 | 76.44 | 49.45 |
| rGAT | **49.98** | **59.70** | **45.30** | 28.92 | 13.24 | 71.56 | 30.70 |
| Wnet2vec | - | - | - | 41.56 | 20.62 | _59.10_ | _26.29_ |

Table 2: KBE models' performance on WN18RR, SimLex999, BATS, PTB POS-tagging, and NER. For SimLex999 and BATS, only subsets of the datasets are used (782 out of 999 word pairs for SimLex999; 17,109 out of 98,000 analogy questions for BATS). Best results are marked in bold, and the lowest ones are underlined.

The choice of SimLex999 is motivated by two factors. Firstly, As opposed to other datasets, it is designed to measure specifically the similarity of concepts, while excluding other forms of relatedness that we would not expect WordNet to capture (e.g. *cup* and *coffee* are related but not semantically similar). Secondly, the dataset focuses on common words, ensuring that representations derived from WordNet have good coverage.

### 4.2 Word Analogy

The Word Analogy task (Mikolov et al., 2013) is designed to measure the linguistic regularities captured within the semantic space of word representations. Specifically, models are tested to predict analogies, i.e. to answer questions of the form $a$ is to $a^*$ as $b$ is to ? (e.g., man is to king as woman is to queen). The underlying assumption is that a high-quality semantic model with a regular latent space should be able to accurately answer these by leveraging the geometric structures embedded within the space (e.g., $v_{a^*} - v_a = v_{b^*} - v_b$). We employ The Bigger Analogy Test Set (BATS) (Gladkova et al., 2016), which categorizes analogy questions into four main categories: Inflection, Derivation, Lexicography, and Encyclopedia. Each category is further divided into ten sub-categories, resulting in a total of 98,000 questions.

### 4.3 Downstream tasks

Another way to understand if embeddings capture semantic knowledge is to assess their performance on downstream tasks. As demonstrated by Chiu et al. (2016), intrinsic semantic measures and downstream tasks do not necessarily correlate. We also include experiments on two sequence la-

belling tasks: POS-tagging, using the Wall Street Journal sections of Penn Treebank (PTB) (Marcus et al., 1993), and NER of CoNLL'03 shared task data (Tjong Kim Sang and De Meulder, 2003).

It is important to note that our objective is not to develop state-of-the-art sequence labelling models, but rather to assess the semantic value of KBE models through comparison. To facilitate a fair evaluation of their semantic capabilities, we employ a minimal setup for the sequence labelling tasks. This involves using a model with a single softmax layer applied to a sliding window. Although this model may yield subpar performance in the tasks, it allows for a reliable comparison between the semantics capabilities of different KBE models. Our setup can be seen as a form of *probing classifier*, testing the degree to which semantic embeddings directly contain the necessary information to predict part-of-speech (Belinkov, 2022).

### 4.4 From Synsets to Words

Nodes in WordNet represent *synsets*, not *words*. As such, to evaluate embeddings of these nodes on our benchmarks, we first convert the synset embeddings obtained from WordNet into word embeddings. This conversion is accomplished by averaging the sense (synset) embeddings corresponding to each word. For instance, the word "car" has two different synsets in WordNet, and the embeddings for these synsets are averaged to obtain the embedding for "car". It is important to note that some words in WordNet appear in multiple synsets with different parts of speech. The word "dog" serves as an example, acting as both a noun representing animals in the Canidae family and a verb meaning to chase. In such cases, we do not dif-

ferentiate the words by part of speech. We base this decision on two reasons: (1) the chosen semantics benchmark are designed for distributional word embeddings, which do not incorporate part of speech information during evaluation, (2) part of speech information is not provided in the BATS word analogy benchmarks. For GNN models (i.e., KBGAT (Nathani et al., 2019) and rGAT (Chen et al., 2021)), which consist of a GNN encoder and a decoder, we use the synset embeddings directly output by the encoder as "embeddings".

## 5 Many Ways to Skin a Potato: Comparing Evaluation Metrics

In this section, we evaluate a range of models (see §2) on both link prediction and also semantic tasks (see §4).

### 5.1 Models trained on WN18RR

The most common link prediction dataset for WordNet is WN18RR, a subset developed by Dettmers et al. (2018). The subgraph represented by WN18RR has been extensively filtered, removing e.g. explicit inverse relations, to produce an excellent link prediction behcnmark. The dataset consists of 93,003 triples with 40,943 entities and 11 relation types. To document the KBE models most likely to appear "in the wild", we begin by evaluating embeddings trained on WN18RR.

As the WN18RR training set only contains a portion of the synsets in WordNet, some words in the semantics benchmark cannot be learned. For BATS, quadruplets containing "missing words" are simply dropped, and the hits-at-10 score on the remaining dataset is reported. For SimLex-999, word pairs containing these "missing words" are assigned the median similarity score obtained from the other word pairs. Despite utilizing subsets of the benchmarks, it's crucial to note that all models are assessed on the exact same selected subsets, ensuring a consistent and fair comparison between them. In terms of Knowledge Base Completion, we use the evaluation protocol defined by Bordes et al. (2013), where the results of the filtered ranking metrics are reported as the mean value of head node and tail node predictions.

Table 2 presents the performance of various KBE models trained on WN18RR, assessed based on a variety of intrinsic and extrinsic tasks. The results allow us to draw two key observations regarding the correlation of metrics. Firstly, a distinct correlation

is evident among the selected semantic tasks; models that excel in one semantic task tend to demonstrate similar prowess across others. Secondly, no clear correlation exists between a model's accuracy in the link prediction benchmark and its efficacy on the semantic tasks. For example, rGAT, despite achieving the highest accuracy on the WN18RR benchmark, underperforms in areas such as similarity, word analogy, POS-tagging, and NER.

The models that showcase top-tier performance on downstream tasks include TransE, FuncE, and KBGAT. A shared characteristic among these models is their reliance on scoring functions motivated by linguistic insights. Both TransE and KBGAT leverage a translational scoring function as defined by (Bordes et al., 2013), which demonstrates a strong geometric association with information-theoretic structure first discovered in distributional word embeddings (Allen et al., 2021). FuncE distinguishes itself by using categorized loss functions derived from the insights of formal semantics (Emerson, 2018, 2020). This stands in stark contrast to models such as rGAT and DistMult, which lack the underpinning of similar linguistic theories. This discrepancy provides a plausible explanation for their inferior performance on the chosen semantic tasks.

### 5.2 Expanded Training

A possible explanation for the unsatisfying performance of KBE models on semantic tasks is the heavily filtered subset that is used for training. For WN18RR, triples are "lost" in three ways: by only focusing on the 18 most frequent relations (Bordes et al., 2013), by removing relations with explicit inverses (Dettmers et al., 2018), and by setting aside triples for the development and test splits. To investigate, we expand the WN18 subgraph to include a more comprehensive set of synsets in WordNet. This expansion occurs in two steps: (1) We extend WN18 by incorporating synsets with low in/out degrees while maintaining the same number of relation types. This expanded subgraph contains 112,195 nodes and 217,495 edges, which is referred to as WN18A. (2) We further extend WN18A with seven additional relations and their corresponding linked synsets, resulting in a dataset with 116,744 nodes and 363,593 edges, referred to as WN25.

Results on the extended datasets are shown in Table 3. For training, we retained hyperparameters from 2 without carrying out hyperparameter

| | WN18A | | | | WN25 | | | |
|---|---|---|---|---|---|---|---|---|
| | Similarity | Analogy | POS | NER | Similarity | Analogy | POS | NER |
| TransE | 58.00 | **29.20** | 79.10 | 48.64 | **50.68** | **36.86** | **80.05** | 47.22 |
| DistMult | 28.32 | 5.80 | 68.58 | 40.39 | 21.82 | 6.35 | 68.57 | 39.57 |
| FuncE | **59.88** | 28.37 | 78.38 | **49.07** | 49.37 | 31.77 | 78.56 | **49.24** |
| MuRP | 31.33 | 15.50 | 74.93 | 40.89 | 23.22 | 28.21 | 76.34 | 38.68 |
| rGAT | 38.53 | 2.94 | 66.84 | 23.65 | 36.35 | 1.66 | 66.77 | 25.52 |
| KBGAT | 58.32 | 29.10 | **79.33** | 48.91 | 48.52 | 35.57 | 79.63 | 47.34 |
| Wnet2vec | 47.16 | 14.08 | 56.09 | 22.78 | 37.08 | 13.82 | 55.32 | 14.77 |

Table 3: KBE models' performance on the semantics benchmarks, with expanded training sets. For SimLex, all the word pairs are covered, while for BATS, 27,343 out of 98,000 quadruplets are covered. Best results are marked in bold, and the lowest ones are underlined.

tuning. This decision was grounded in the fact that both WN18A and WN25 have no validation set, so carrying out hyperparameter tuning would result in over-fitting. Notably, the results exhibit a similar pattern to those in Table 2: rGAT and DistMult consistently perform poorly across all semantic tasks, while models with scoring functions designed with linguistic motivations demonstrate better performance; in other words, there is still an inverse correlation between link prediction and downstream results.

It is important to note that some of the results in Table 3 are not directly comparable to Table 2, as they have different coverage on the benchmarks. This is particularly evident in the word analogy task, where the ranking metric is strongly influenced by the total number of nodes in the graph. Interestingly, the performance of rGAT and DistMult is less improved by the dataset extension compared to other KBE models – these models exhibit sub-par performance on word analogy tasks and show no improvement in the POS tagging benchmark. This suggests that the architecture, rather than e.g. a lack of training data, is the central problem for these models.

## 6 WordNet Potatoes and KBE Model Architectures: A Problematic Mash

As a concrete demonstration of how KBE models fail to accurately represent semantics, we consider the case of hypernymy. WordNet encodes a tree defined by the hyponymy and hypernymy relations, with concepts near the leaves representing more concrete forms of those occurring near the root. This is a transitive relation: A *potato* is a concrete example of a *vegetable*, and therefore also a con-

crete example of *food*. To accurately model the semantic information in WordNet, models must encode this hierarchical structure in their latent spaces. Unfortunately, models are typically not designed for this. We find that some of our selected models exhibit unexpected behavior when encoding hypernymy, and almost all of them fail to properly capture the transitive property.

### 6.1 Model Behaviour Contradicts Thesaurus-Based Similarity Measures

Consider an upward path in the hypernym hierarchy of WordNet (e.g., potato → solanaceous vegetable → produce → food → solid → matter → physical entity). It is reasonable to expect that concepts closer to each other in the path would have higher similarity scores. This approach is commonly used in thesaurus-based similarity algorithms (Quillian, 1969). However, KBE models are not trained to exhibit this property. Table 4a presents the cosine similarity between potato and all of its hypernyms. We observe that TransE and FuncE strictly follow the thesaurus-based similarity idea, with solanaceous vegetable being the most similar to potato and physical entity being the least similar. In contrast, DistMult and rGAT give similarity scores that are not clearly correlated with the path length.

The differing behaviour of models on this task reflect their differing motivation. As mentioned in Section 5.1, both TransE and FuncE are theoretically motivated, respectively by ideas from the geometry of word embeddings (Allen et al., 2021) and from formal semantics (Emerson, 2018, 2020). DistMult and rGAT, on the other hand, are primarily motivated as being *good models for link prediction*. While they accomplish the latter goal

| Cosine Similarity with 'Potato' | | | |
| --- | --- | --- | --- |
| | TransE | DistMult | rGAT |
| solanaceous veg. | 0.7750 | −0.0009 | 0.6927 |
| vegetable | 0.5468 | 0.6390 | 0.5026 |
| produce | 0.2723 | −0.3153 | 0.3991 |
| food | 0.1676 | 0.4571 | 0.3481 |
| solid | 0.0623 | −0.3725 | 0.1821 |
| matter | 0.0601 | 0.3959 | 0.2130 |
| physical entity | 0.0236 | −0.3699 | 0.4192 |

(a) Cosine similarity between 'Potato' and all its hypernyms in WordNet.

| Hypernym Prediction @1 | | | | |
| --- | --- | --- | --- | --- |
| | TransE | DistMult | FuncE | MuRP |
| 1-hop | 84.41 | 83.96 | 86.32 | 83.45 |
| 2-hop | 2.21 | 10.08 | 76.65 | 9.73 |
| 3-hop | 0.81 | 4.44 | 47.5 | 4.03 |
| 4-hop | 0.58 | 0.41 | 10.2 | 1.56 |
| 5-hop | 0.56 | 0.12 | 3.76 | 0.53 |
| 6-hop | 0.38 | 0.04 | 1.68 | 0.32 |
| 7-hop | 0.30 | 0.01 | 0.84 | 0.11 |

(b) Predicting multi-hop hypernyms. KBEs are trained and tested on WN18A.

Table 4: Investigation of KBEs' ability to model hyponymy and hypernymy relations.

| | WN18RR | | | Similarity | Analogy | POS tagging | NER |
| --- | --- | --- | --- | --- | --- | --- | --- |
| | MRR | @10 | @1 | @10 | Spearman $\rho$ | acc. | f1 score |
| KBGAT-TransE | 26.03 | 48.64 | 20.44 | 45.05 | 31.56 | 76.44 | 47.36 |
| KBGAT-FuncE | 23.87 | 42.52 | 15.03 | 48.77 | 25.60 | 77.03 | 52.12 |
| rGAT-TransE | 22.43 | 48.03 | 17.31 | 46.76 | 28.60 | 76.92 | 48.55 |
| rGAT-FuncE | 14.45 | 39.17 | 9.56 | 47.89 | 30.60 | 77.22 | 51.92 |

Table 5: Results for GNNs with modified scoring functions, to compare against Table 2.

(e.g. in terms of MRR on WN18RR, rGAT performs the best out of all the models we test), they in turn sacrifice the ability to model the underlying semantics of WordNet. This behaviour is also reflected through the comparatively poor performance of these models on our semantic tasks (see Tables 2 and 3).

## 6.2 Modelling Transitivity

The hypernymy relation in WordNet is transitive: if $(a, \text{hypernym}, b)$ and $(b, \text{hypernym}, c)$ are valid triplets, then $(a, \text{hypernym}, c)$ is also valid. However, almost none of the KBE models proposed are capable of modelling transitivity. For instance, TransE is trained to enforce that $v_s + v_r \approx v_o$ if $(s, r, o)$ is a valid triplet. This method is deficient for modelling transitive relations, since $(v_a + v_r \approx v_b) \wedge (v_b + v_r \approx v_c)$ can only imply $(v_a + v_r \approx v_c)$ if $v_r \approx \mathbf{0}$. Similarly, bilinear scoring functions and neural decoders do not easily allow the representation of transitivity.

Table 4b demonstrates the average accuracy for different models in predicting whether $n$-hop hypernymy (e.g. predicting whether a potato is a vegetable, whether it is produce, whether it is food, and so on). As can be seen, TransE and DistMult

fail to predict multi-hop hypernymy. FuncE, which is explicitly designed to model transitivity, is to a much greater degree able represent hypernyms further away.

Drawing from these observations, we emphasize the significant potential inherent in specialized KBE models tailored for specific kinds of data. We urge future model designers to meticulously consider the desired functionalities of their models, ensure that they are able to achieve these functionalities in theory, and evaluate whether they achieve them in practice.

## 6.3 Scoring Functions for GNN-encoders

GNN encoders, e.g. rGAT (Chen et al., 2021), are often designed to work with neural scoring functions. As these are designed to maximize link prediction performance, rather than capture the latent structure of the semantic space, their corresponding embeddings may not work well for downstream tasks. A possible remedy would be to pair these powerful encoders with linguistically motivated decoders (e.g., TransE or FuncE).

We test this proposal in Table 5, pairing rGAT and KBGAT with TransE- and FuncE-decoders. We find that, although performance increases for

| | | Word Similarity | | | Word Analogy | | |
|---|---|---|---|---|---|---|---|
| | | Adj. | Noun | Verb | Deri. | Lexi. | Ency. |
| WN18A | TransE | 70.30 | 53.32 | 62.74 | 44.12 | 28.50 | 17.45 |
| | DistMult | 37.80 | 27.07 | 28.32 | 13.97 | 2.84 | 1.39 |
| | FuncE | **71.10** | **53.80** | **67.76** | 39.40 | 28.92 | 15.23 |
| | MuRP | 29.83 | 34.39 | 42.15 | 24.31 | 12.52 | 11.41 |
| | rGAT | 44.04 | 35.99 | 46.91 | 0.26 | 3.88 | 5.01 |
| | KBGAT | 70.45 | 52.98 | 63.70 | 43.78 | 28.52 | 16.70 |
| | Wnet2vec | 64.40 | 43.13 | 46.88 | 16.51 | 15.40 | 9.63 |
| WN25 | TransE | 55.45 | 49.66 | 47.22 | **47.40** | **38.86** | **19.87** |
| | DistMult | 30.35 | 23.15 | 12.76 | 12.51 | 4.03 | 1.92 |
| | FuncE | 52.64 | 50.99 | 47.09 | 44.95 | 34.82 | 15.92 |
| | MuRP | 30.58 | 39.85 | 28.04 | 23.85 | 15.97 | 12.56 |
| | rGAT | 31.17 | 40.26 | 29.88 | 0.27 | 2.64 | 2.18 |
| | KBGAT | 54.52 | 48.23 | 46.81 | 45.99 | 38.21 | 18.60 |
| | Wnet2vec | 41.67 | 40.94 | 32.51 | 22.88 | 18.29 | 9.76 |

Table 6: Results on SimLex999 and BATS, by category (part of speech and derivational/lexicographic/encyclopedic). Best results are marked in bold, and the lowest ones are underlined.

semantic tasks, there is a corresponding drop in MRR; further, the addition of the GNN-encoder to TransE or FuncE does not improve on the decoder-only variant (see Table 2). This matches existing work pairing GNN encoders with translational scoring functions (Cai et al., 2019), which also failed to yield significant improvements over using the factorization model by itself. Indeed, the majority of GNN architectures proposed for link prediction use either bilinear (Schlichtkrull et al., 2018) or neural (Nathani et al., 2019; Chen et al., 2021) scoring functions. As we discuss in Section 6.1, such scoring functions do not represent the hierarchical structure of WordNet well. Our findings suggest that the development of a GNN encoder which pairs well with linguistically motivated scoring functions – or a linguistically motivated scoring function which pairs well with GNNs – is a highly desirable direction for future research.

## 7  Further Analysis: Subtasks Under the Spudlight

An interesting question is how uniform our findings are across various subtasks of our evaluation tasks. Conveniently, SimLex-999 and BATs allow for such analysis, respectively across POS-tag and question category. Detailed model results for these can be seen in Table 6.

On SimLex-999, a fairly consistent pattern emerges: models perform better on adjectives than on nouns or verbs. While present for both datasets, this trend is more pronounced for WN18A than WN25. One possible interpretation is that KBE models tend to attribute high similarity scores to concepts that are highly connected in the training data, i.e. to (potential) edges between vertices in denser regions of WordNet. This leads to heightened similarity scores between concepts that share strong associations, despite having less similarity; contradicting the measurement principles of Sim-Lex (Hill et al., 2015b). This is particularly prominent for noun synsets, as the majority of connections in WordNet occur between nouns. In contrast, there are no connections between adjectives in WN18A, and only a few connections in WN25.

Analogy questions in BATS are classified into four groups according to the original paper: *derivational*, *lexicographical*, *encyclopedic*, and *inflectional*. We exclude the inflectional group due to its limited coverage in WordNet. As per Table 6, KBE models tend to perform the best in the derivational group. This outcome is unexpected, given that a large number of analogy questions in the lexicographical group can be answered using relations directly from WordNet (e.g., analogy questions involving hypernym-hyponym, part-whole, and meronym-holonym). In contrast, derivational relations in WordNet are encoded in a more general

manner, utilizing a single 'derivationally related' relation. Moreover, the performance of KBE models diverges from the trend observed in distributional word embeddings, which typically underperform in the derivational group (Gladkova et al., 2016).

For both datasets, the ranking of the models by performance is broadly consistent between the data subsets. The impact of the additional relations in WN25 is inconsistent, generally improving performance on BATS but decreasing it on SimLex999.

## 8 Conclusion

Our study has highlighted the substantial discrepancy between the performance of KBE models on the WN18RR benchmark versus their performance in semantics and downstream tasks. Specifically, we have shown that KBE models, when designed absent of linguistic insights, are capable of delivering high performance on link prediction metrics while also underperforming and demonstrating unexpected behaviors in semantic analysis. Our findings demonstrate the deficiency of the WN18RR benchmark in evaluating specific semantic capabilities of KBE models and underscore the potential advantages of constructing models tailored for semantic tasks. Consequently, we advocate for the development of targeted benchmarks that assess specific semantic abilities of WordNet KBE models, which enable model designers to calibrate and optimize their models in alignment with precise semantic requisites.

## Limitations

Our evaluation of the semantic capabilities of KBE models is limited in several ways. We focus on specific aspects of semantics, and two downstream tasks. The semantic tasks we have included are not perfect representations of semantic evaluation, and indeed do not always reflect downstream performance (Chiu et al., 2016). Further, POS tagging and NER are only two of many options for downstream tasks requiring some semantic knowledge. As such, it is possible that KBE evaluation accurately captures *other* aspects of semantics. Additionally, we note that our analysis is restricted to one ontology, WordNet, which we chose because it is often used as a benchmark for KBE models. A different ontology, e.g. BabelNet (Navigli and Ponzetto, 2010), may produce different results.

We further note that the models we use in this paper, i.e. for semantic tasks and for POS tagging, are purposefully made simple rather than performant. Our intention was not to develop state-of-the-art systems for these tasks, but rather to analyse the embeddings produced by relational link prediction.

## Acknowledgements

We extend our deepest gratitude to Charles Chen for his pioneering undergraduate thesis, which laid groundwork for the FuncE model. We are also thankful to Tiago Pimentel, Rami Aly, and the anonymous reviewers, whose insightful feedback and constructive criticism was instrumental in refining our manuscript to its present form.

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

## A    Relation Categories

| Relation Categories | |
| --- | --- |
| Hypernym-based | **hyponym** 
 **instance hyponym** 
 hypernym 
 instance hypernym |
| Synonym-based | **verb group** 
 **similar to** 
 **also see** 
 **derivationally related form** |
| Other | **member meronym** 
 **has part** 
 **member of domain region** 
 **member of domain usage** 
 **synset domain topic of** 
 member holonym 
 part of 
 synset domain region of 
 synset domain usage of 
 member of domain topic |
| Other (WN25) | attribute 
 cause 
 entailment 
 pertainym 
 antonyms 
 substance holonym 
 substance meronym |

Table 7: WordNet relations grouped into categories. The first 18 relations are used in WN18. The 11 relations used in WN18RR are marked in bold.

## B  Hyper-parameters

|          | learning algorithm | learning rate | batch size | dimension | epoch | weight decay |
|----------|--------------------|---------------|------------|-----------|-------|--------------|
| TransE   | Adam               | 1e-2          | 16384      | 200       | 400   | 1e-2         |
| DistMult | Adagrad            | 5e-1          | 8192       | 200       | 400   | 1e-1         |
| FuncE    | Adam               | 1e-2          | 8192       | 200       | 400   | 1e-1         |
| MuRP     | RiemannianSGD      | 50            | 128        | 200       | 200   | 0            |
| rGAT     | Adam               | 1e-3          | 4096       | 200       | 800   | 1e-5         |
| KBGAT    | Adam               | 1e-3          | 4086       | 200       | 3600  | 1e-5         |
| Wnet2vec | -                  | -             | -          | 800       | -     | -            |

Table 8: Hyper-parameter choices for model trained on WN18RR, WN18A, and WN25

## C  Benchmark Examples

| Benchmark      | Sample |
|----------------|--------|
| SimLex999      | *Word Pair:* (old, new), *Similarity Score:* 1.58 |
| BATS           | *Analogy question:* man : woman :: king : ? 
 *Answer:* queen |
| PTB POS tagging | *Sentence:* The quick brown fox jumps over the lazy dog. 
 *Tags:* DT JJ JJ NN VBZ IN DT JJ NN. |
| CoNLL-2003 NER | *Sentence:* Mike lives in London. 
 *Tags:* [Mike, B-PER] [lives, O] [in, O] [London, B-LOC] |

Table 9: Examples from the used benchmarks