# OpenReview forum: "Are Embedded Potatoes Still Vegetables? On the Limitations of WordNet Embeddings for Lexical Semantics"
_EMNLP/2023/Conference — EMNLP 2023 Main_

### Official Review · Reviewer_UJC6 · 2023-07-26

**Soundness:** 3

**Excitement:**

4: Strong: This paper deepens the understanding of some phenomenon or lowers the barriers to an existing research direction.

**Paper Topic And Main Contributions:**

The paper is on a limitation of knowledge graph embeddings: while performance in tasks like link prediction can be good, concepts often lose their semantic properties.
This is confirmed by experiments performed in a link prediction dataset and in two "semantics" tasks: similarity and analogy.
Top performances in WN18RR, a link prediction dataset, are obtained by embeddings that perform poorly in the semantic tasks. On the other hand, FuncE, an introduced method that uses different scoring functions for each relation, performs better in the semantic tasks and not so well in link prediction.

**Questions For The Authors:**

Question A: The maths for reaching 99,200 questions in BATS should be clarified, starting with the number of entries per sub-category and how they are combined. Is this number still true after excluding the inflection category? Also, does it make sense to include the derivational category when testing semantics?

Question B: Why are the reported results computed on a subset of the datasets? If it is due to coverage, is coverage the same for all methods? How does this affect the conclusions?

Question C: Is it ok to merge different word senses of the same word in the same node and simply averaging the different sense embeddings?

Question D: What were the hyperparameters for the results in tables 2 and 3? Are they not important?

Question E: Could you provide more information on pairing the GNN encoders with the linguistically-motivated decoders?

**Reasons To Accept:**

The paper has a very interesting premise, rare in papers that we see published nowadays, followed by an experimentation designed for confirming the previous.

**Reasons To Reject:**

Here and there, the paper could be better organised and clearer. For instance, it would be easier to follow if it included a better description of the tasks and datasets, also with examples of their content.

It is also not clear why the reported results are on subsets of the datasets, and how this impacts the conclusions.

As the authors mention in the Limitations section, some conclusions can be limited by the number of models and tasks tackled.

**Reproducibility:**

3: Could reproduce the results with some difficulty. The settings of parameters are underspecified or subjectively determined; the training/evaluation data are not widely available.

**Reviewer Confidence:**

4: Quite sure. I tried to check the important points carefully. It's unlikely, though conceivable, that I missed something that should affect my ratings.

**Typos Grammar Style And Presentation Improvements:**

Table 1 appears much earlier than mentioned in text, and it mentions Table 2, which is even further ahead, which makes their interpretation difficult.

The structure of the paper is not provided in the Introduction.

GNN is used before appearing in full.
Names of the computed metrics are never in full.

WN18 is first mentioned without its contents being described.

---

> ### Author Rebuttal · Authors · 2023-08-29
>
> We thank the reviewer for their very positive comments. We are delighted to hear that they find our premise uniquely interesting among NLP papers – we appreciate the support!
>
> **Organisation & Clarity**
>
> We thank the reviewer for pointing this out. In the camera ready, we will include an example from each dataset to better illustrate the tasks we evaluate on.
>
> **Subsets**
>
> The models we compare in this paper are all trained on (various subsets of) WordNet. To minimise noise in our evaluation, we evaluated on the subsets of SimLex and BATS covered by WordNet. Coverage, training, and evaluation was the same for all methods (trained on a WordNet fragment, evaluated on the part of SimLex/BATS covered by WordNet).
>
> Using the entire evaluation datasets would not change our findings – the datasets would just be padded with examples where all methods (being trained on WordNet) would perform at chance; hence, added noise. We will clarify this in the camera ready.
>
> Regarding BATS, 99,200 is the number of examples in the original dataset (Gladkova et al., 2016). After filtering out examples not included in the training set, we are left with 17,109 (WN18RR training set, see caption for Table 2) or 27,343 (full training set, see caption for Table 3). We will also make this clearer in the camera ready, moving the relevant details into the main text of Section 4.2. As WordNet uses lemmas rather than inflected wordforms, the inflected category is almost entirely filtered out. Many derivationally related words are included in WordNet, even if the morphological structure is not indicated. Derivationally related words have semantic relationships, and as seen in Table 7, WordNet-trained models generally perform better on this category than on the other BATS categories.
>
> **Merging Senses**
>
> We agree that merging all senses by averaging sense embeddings into a single word embedding would not be optimal, if the goal was performance. Probabilistically, this strategy can be seen as assuming a uniform distribution (i.e. equal likelihood) over all senses of a word, and marginalising out the senses. Other distributions might have led to higher numbers – however, as long as we use the *same* marginalisation for all systems, this would not impact the correlations between results on KBE datasets and semantic or downstream tasks.
>
> **Hyperparameters**
>
> We thank the reviewer for spotting this – we will include all hyperparameters in the camera-ready in an appendix. We are also making our source code available, so that all experiments will be reproducible. The important point about using the same hyperparameters between Tables 2 and 3 is methodological: for Table 2, the WN18RR dataset is split between training, validation, and test, and so hyperparameters can be tuned on the validation set; for Table 3, the training set is expanded to include all of WordNet so there is no validation set. It would be methodologically problematic to tune hyperparameters on the evaluation datasets in Table 3, and so we used the same hyperparameter values as in Table 2. We can clarify this methodological point in the camera-ready.
>
> **Linguistically Motivated GNN Decoders**
>
> The GNN encoders we experimented with – KBGAT, rGAT – did not, from our findings, function well when using e.g. FuncE or TransE as the decoder. As the decoder (or scoring function) appears to limit quite heavily what performance can be expected, and what phenomena can be modelled, this is a serious limitation of GNN-based KBE models. In order to use GNNs to model semantics (at least in the form of KBE’s of WordNet), this limitation would need to be overcome. This would likely involve an investigation of why some decoders (DistMult) work so much better with GNNs than others (TransE), and potentially the development of new GNNs or decoders that solve the issue. These experiments, while interesting, are a bit outside the scope of our paper – but, we think, a very fruitful direction for future research.

---

### Official Review · Reviewer_P9Hk · 2023-08-01

**Soundness:** 4

**Excitement:**

4: Strong: This paper deepens the understanding of some phenomenon or lowers the barriers to an existing research direction.

**Paper Topic And Main Contributions:**

The paper studies the pitfalls of Knowledge Base Embeddings (KBE) learned representations in various settings;
- the actual link prediction task, on which these models are trained to do
- word semantic similarity
- analogy
- and part of speech tagging (POS).

The authors show that link prediction performance does not correlate with semantic performance or the downstream task POS.
Their experiments focus on hierarchical relations between different WordNet entities (such as potato -> vegetable -> food).
It is clear, from the authors' conclusion, that link prediction performance should not be used as a proxy to assess the downstream performance of KBE.

I would like to add that this paper is as thorough as self-contained, and is really easy to read and understand.

**Questions For The Authors:**

- More of a philosophical question regarding the very root motivation of your paper: Why would we use KBE models for word similarity (to some extent I can see why) or analogy and POS tagging? should we just use these types of model for link prediction? What motivated you to investigate this knowledge transfer in the first place? Maybe models specifically designed to predict hypernymy would be better suited for the task?
- This got me questioned about the usefulness of KBE models if their only usage is for link prediction. Maybe that's their only utility, in the end.

**Reasons To Accept:**

- Thorough analysis of KBE pitfalls with respect to semantic and downstream evaluation which does not correlate with their explicit training objective i.e. link prediction.
- The experiments conducted are well sounded, and every hypothesis/assumption made by the authors is backed by well-formulated arguments.
- This paper compares several KBE models which strengthened their claim

**Reasons To Reject:**

The only weakness I see is that there's only one downstream task i.e. POS tagging. Would NER be a bit more interesting to evaluate?

**Reproducibility:**

4: Could mostly reproduce the results, but there may be some variation because of sample variance or minor variations in their interpretation of the protocol or method.

**Reviewer Confidence:**

4: Quite sure. I tried to check the important points carefully. It's unlikely, though conceivable, that I missed something that should affect my ratings.

**Typos Grammar Style And Presentation Improvements:**

- Page 1 Table 1: H@1 is unfamiliar (at least to me). I would have expected P@1 or P@10. It's only at page 5 line 362 that I understood what H@1 meant.
- Line 353: behcnmark -> benchmark
- Would that be possible to differentiate the models based on a scoring function from the other ones in the tables? That would improve reading/understanding the results
- Table 6: Switch second header of word similarity and word analogy (word sim -> spearman, word analogy -> @10)
- Table 6: Bring table 2 in table 6 so we don't need to do back and forth constently
- Even though I know BATS and SimLex999, I was sometimes wondering what were they coming from when the authors were analyzing the results. would there be a way to harmonize Word Similarity == SimLex99 and Word Analogy == BATS

---

> ### Author Rebuttal · Authors · 2023-08-29
>
> We thank the reviewer for their positive feedback. We are happy they found our paper to be well-formulated, thorough, and easy to read!
>
> **Additional Downstream Task**
>
> Our primary intent with this paper was to investigate the correlation between link prediction and semantic/downstream tasks. We expect embedding performance on NER and POS tagging to correlate with each other, as found in previous work (for example: POS tags are highly informative features for NER tagging [1]; out of a set of 33 NLP tasks, there was a high level of transferability between POS tagging and NER [2]), and as such we would not expect to see substantial differences when using NER instead of POS tagging. With that said, we appreciate the suggestion, and agree that adding another task would provide further evidence. Other tasks could include sentiment analysis or text classification.
>
> [1] A Multi-task Approach for Named Entity Recognition in Social Media Data. Aguilar et al., WNUT, at ACL 2017. https://aclanthology.org/W17-4419/
>
> [2] Exploring and Predicting Transferability across NLP Tasks. Vu et al., EMNLP 2020. https://aclanthology.org/2020.emnlp-main.635/
>
> **KBE Models for Semantics**
>
> We appreciate this excellent question, and we hope that our paper will prompt more researchers to think about these underlying issues. KBE models are often motivated as representations of the underlying structure of the knowledge base. If the knowledge base is WordNet, a KBE model should therefore encode the structure of WordNet – things like hypernymy, meronymy, and so on. The intuition is that this knowledge should be helpful for more general semantic modelling. These are implicit assumptions; our aim is to test them.
>
> On a more practical level, KBEs are used in many applications. We mention a few in the introduction, on lines 27-30: entity typing (Moon et al.,027 2017; Zhao et al., 2020), entity alignment (Sun et al., 2020), rule mining (Yang et al., 2015; Chen et al., 2020), and conceptual clustering (Gad-Elrab et al., 2020).
>
> Our findings suggest that the usefulness of KBEs beyond link prediction is difficult to estimate based on link prediction performance. This doesn’t mean that KBE models shouldn’t be used on other tasks, and in fact even our worst-performing models still perform better than chance (and Wnet2vec, which is not trained as a link prediction model).  However, other metrics are necessary to select the best KBE models for semantic and downstream tasks, and the past years of selecting models by SOTA performance on KBE benchmarks would not have revealed them!

---

### Official Review · Reviewer_YVuz · 2023-08-06

**Soundness:** 3

**Excitement:**

4: Strong: This paper deepens the understanding of some phenomenon or lowers the barriers to an existing research direction.

**Paper Topic And Main Contributions:**

The paper provides a comprehensive study of a large amount of KBE models on link prediction and their ability to perform in semantic analysis tasks. The authors show a negative correlation between performance on the WN18RR benchmark of KBE models with two semantic analysis tasks, and one downstream task. These results demonstrate the need for careful design of both evaluation benchmarks and model architectures.

**Questions For The Authors:**

N/A

**Reasons To Accept:**

- The paper evaluates a lot of state-of-the-art KBE models, which show a clear negative correlation between different tasks.
- The findings in the paper are very interesting and important to further research and real-life application.
- The authors show a clear and insightful discussion of the main results and findings, which allows to clearly show the problem.


**Reasons To Reject:**

- Evaluation of the downstream task is very limited, it could be useful to present a more challenging downstream task, than POS tagging.
- The authors do not discuss how to further solve this issue.


**Reproducibility:**

4: Could mostly reproduce the results, but there may be some variation because of sample variance or minor variations in their interpretation of the protocol or method.

**Reviewer Confidence:**

3: Pretty sure, but there's a chance I missed something. Although I have a good feel for this area in general, I did not carefully check the paper's details, e.g., the math, experimental design, or novelty.

---

> ### Author Rebuttal · Authors · 2023-08-29
>
> We thank the reviewer for their comments. We are happy they found our paper comprehensive, interesting, and important!
>
> **Additional Downstream Task**
>
> Our primary intent with this paper was to investigate the correlation between link prediction and semantic/downstream tasks. As suggested by reviewer 2, NER tagging would be another option for evaluation. However, we expect embedding performance on NER and POS tagging to correlate with each other, as found in previous work (for example: POS tags are highly informative features for NER tagging [1]; out of a set of 33 NLP tasks, there was a high level of transferability between POS tagging and NER [2]), and as such we would not expect to see substantial differences when using NER instead of POS tagging. With that said, we appreciate the suggestion, and agree that adding another task would provide further evidence. Other tasks could include sentiment analysis or text classification.
>
> [1] A Multi-task Approach for Named Entity Recognition in Social Media Data. Aguilar et al., WNUT, at ACL 2017. https://aclanthology.org/W17-4419/
>
> [2] Exploring and Predicting Transferability across NLP Tasks. Vu et al., EMNLP 2020. https://aclanthology.org/2020.emnlp-main.635/
>
> **Possible Solutions**
>
> We include several quite general recommendations in the paper, i.e. on lines 100-102: *“evaluation must target desired functionality, and models must be theoretically capable of that functionality”*. If we were to theorise more specific suggestions, they would fall along these lines as well. For example, section 6.2 looks specifically at hypernymy – we can expand this discussion in the camera-ready.
>
> For evaluation, our primary suggested solution is for researchers to not use link prediction benchmarks to estimate performance on semantic or downstream tasks – instead, we recommend directly evaluating on the target task. That way, low correlation between metrics is not an issue. We emphasise that link prediction might still be a useful training task, though.
>
> For modelling, we recommend thinking carefully about what functionalities a system should be capable of, and what the “shape” of the target task is. In our experiments, we showed that FuncE – a relatively simple model that incorporates *different, hand-crafted* scoring functions for different relation types performs very well on semantic and downstream tasks. We see this as a fruitful direction: specialist models designed for the target task.
>
> We will expand our discussion of both of these points in the camera ready.

---

### Meta-Review · Area_Chair_PoNR · 2023-09-13

**Recommendation:** 5

**Metareview:**

The paper tackles the common approach to evaluating of Knowledge-Base Embeddings (KBE) on link prediction task. It shows, comparing several KBE models that performance in this task does not correlate with performance in the downstream task (POS tagging in this case).
All three reviewers agreed that the paper is sound and interesting. The main objection was that the evaluation is performed using only one downstream task, namely POS tagging. During the rebuttal period the authors conducted additional experiments and showed that they support the main claim of the paper. I think these results make the paper stronger, and should be reported in the final version.
Other objections concerned main assumptions and positioning of the paper. I found the authors' responses convincing, and they probably should be reproduced on the extra page.

---

### Decision · Program_Chairs · 2023-10-07

**Decision:**

Accept-Main

**Comment:**

The paper tackles the common approach to evaluating of Knowledge-Base Embeddings (KBE) on link prediction task. It shows, comparing several KBE models that performance in this task does not correlate with performance in the downstream task (POS tagging in this case).
All three reviewers agreed that the paper is sound and interesting. The main objection was that the evaluation is performed using only one downstream task, namely POS tagging. During the rebuttal period the authors conducted additional experiments and showed that they support the main claim of the paper. I think these results make the paper stronger, and should be reported in the final version.
Other objections concerned main assumptions and positioning of the paper. I found the authors' responses convincing, and they probably should be reproduced on the extra page.